**Data Availability Statement:** Deidentified data summaries and analysis file are provided in supplemental material. Our audit was conducted

# Management of tuberculosis infection in Victorian children: A retrospective clinical audit of factors affecting treatment completion

**Rebecca Helena Holmes**[1], **Sunjuri Sun**[2], **Saniya Kazi**[3], **Sarath Ranganathan**[4,5], **Shidan Tosif**[4,5,6], **Stephen M. Graham**[4,5,6], **Hamish R. Graham**[4,5,6]*

1 Melbourne Medical School, University of Melbourne, Melbourne, Australia, 2 Nuffield Department of Population Health, University of Oxford, Oxford, United Kingdom, 3 Department of Paediatrics, Monash Health, Clayton, Australia, 4 Department of Respiratory Medicine, Royal Children's Hospital, Melbourne, Australia, 5 Department of Paediatrics, University of Melbourne, Melbourne, Australia, 6 Infection and Immunity, Murdoch Children Research Institute, Melbourne, Australia

* Hamish.Graham@rch.org.au

## Abstract

### Background

Tuberculosis preventive treatment (TPT) is strongly recommended for children following infection with *Mycobacterium tuberculosis* because of their high risk of progression to active tuberculosis, including severe disseminated disease. We describe the implementation of TPT for children and adolescents with evidence of tuberculosis infection (TBI) at Victoria's largest children's hospital and examine factors affecting treatment completion.

### Methods

We conducted a retrospective clinical audit of all children and adolescents aged <18 years diagnosed with latent TBI at the Royal Children's Hospital, Melbourne, between 2010 and 2016 inclusive. The primary outcome was treatment completion, defined as completing TPT to within one month of a target duration for the specified regimen (for instance, at least five months of a six-month isoniazid course), confirmed by the treating clinician. Factors associated with treatment adherence were evaluated by univariate and multivariate analysis.

### Results

Of 402 participants with TBI, 296 (74%) met the criteria for treatment "complete". The most common TPT regimen was six months of daily isoniazid (377, 94%). On multivariate logistic regression analysis, treatment completion was more likely among children and adolescents who had refugee health screening performed (OR 2.31, 95%CI 1.34–4.00) or who were also treated for other medical conditions (OR 1.67 95%CI 1.0–2.85), and less likely among those who experienced side-effects (OR 0.32, 95%CI 0.11–0.94). However, TPT was generally well tolerated with side-effects reported in 15 participants (3.7%).

under the Royal Children's Hospital permissions for quality improvement activities and does not include permission to share individual patient data in an open access repository. The data underlying the results presented in the study are available for researchers who meet the criteria for access to confidential data from The Royal Children's Hospital HREC (rch.ethics@rch.org.au; +61 393455044).

**Funding:** HG and SR received a grant from the John Burge Trust Fund (no specific grant number) including salary support for SS. The funder did not play any role in the study design, data collection and analysis, decision to publish, or preparation of the manuscript https://www.statetrustees.com.au/philanthropy-and-charitable-giving/granting/tuberculosis.

**Competing interests:** HG, SG, ST, SK, SR are employed by the Royal Children's Hospital and work in the TB clinic. This does not alter our adherence to PLOS ONE policies on sharing data and materials.

## Conclusion

Identification of factors associated with TPT completion and deficiencies in the existing care pathway have informed service provision changes to further improve outcomes for Victorian children and adolescents with TBI.

## Introduction

Infection with *Mycobacterium tuberculosis* (TBI) represents a major global public health concern, with approximately a quarter of the world's population estimated to have been previously infected and 10 million people acquiring the infection annually [1]. It is estimated that in 2020, there were 1.5 million TB-related deaths worldwide, with 230,000 TB-related deaths in children [1]. The treatment of infection with tuberculosis preventive treatment (TPT) is recommended for young child contacts on the basis of strong evidence of disease risk following recent exposure and of TPT effectiveness [2, 3]. The updated global recommendations for TB-endemic countries now extend to HIV-negative older children and adolescents, a group that is already eligible for TPT in a low-endemic, resource-rich country such as Australia [4]. However, implementation of TPT for older children and adolescents with infection and who are HIV-negative has been negligible in tuberculosis-endemic countries [1].

A major challenge for TPT implementation has been complete adherence to what has traditionally been a TPT regimen that requires daily treatment with isoniazid for at least six months [5, 6]. However, shorter course TPT regimens are now also recommended that have equal effectiveness to isoniazid preventive therapy while having the advantages of being safer and having improved treatment adherence [3, 7]. As such, the updated TPT guidelines in the state of Victoria, Australia now recommend 4 months of rifampicin (4R) for older children and adults and a fixed dose combination of isoniazid-rifampicin for 3 months (3HR) in younger children [4]. Alternative options (or monitoring without TPT) may be considered if there are adverse effects to these medications, or if the index case is known to be drug-resistant.

Overall, there is a lack of data on the factors that influence adherence to/or completion of TPT, particularly in children and in low prevalence settings, and therefore further research is warranted in this area to inform clinical practice and improve treatment outcomes. This project examined the factors affecting treatment completion for children diagnosed with TBI at the Royal Children's Hospital (RCH) in Melbourne. By identifying these factors, we hoped to improve the service to better address the needs of the community and improve treatment completion rates. We hypothesised that individual, social and clinical factors identified in this study (such as age, English Language proficiency and clinic attended) will affect treatment completion rates, and that some of these factors may be modifiable by service providers to deliver TPT more effectively.

## Methods

### Design

We conducted a retrospective clinical audit of all children (aged <18 years old at the time of diagnosis) who were seen at RCH for TBI between 1 January 2010 and 31 December 2016. The study was determined exempt from formal ethics approval as it was a retrospective clinical audit for quality improvement purposes and it did not involve individual patient recruitment or consent. Trained investigators identified potentially eligible patients through outpatient

clinic lists (from the Respiratory Medicine Tuberculosis clinic, General Medical Immigrant Health clinic, and Infectious Diseases clinics) and pharmacy prescriptions were recorded (for all patients dispensed isoniazid or rifampicin). We included patients who were aged less than 18 years old at time of diagnosis, with latent TBI diagnosed by a treating clinician and who had not been previously treated for TBI.

Patient data were obtained from electronic medical records (Epic Systems, Wisconsin, US) using a custom data collection form and input into REDCap (Research Electronic Data Capture [8, 9]) by a single investigator. Data cleaning and analysis was performed using StataSE v.16 (StataCorp, Texas US). Any discrepancies found in the dataset were referred to a senior investigator who cross-checked the medical records and corrected the data.

## Population

As this audit was designed to inform clinical practice, data were collected from the previous seven years, to provide reasonable representation of recent practices. We identified eligible participants using TB-related diagnostic codes, and manual review of patients attending the TB and Immigrant Health clinics and patients prescribed TB medications from RCH pharmacy.

## Variables

The primary outcome was TPT completion, defined as completing pharmacological treatment to within 1 month of a target duration for the specified agent (for instance, at least 5 months of a 6-month isoniazid course), confirmed by the treating clinician. We determined this by identifying the recommended treatment and date of commencement, then checking all subsequent clinical encounters to find clinical documentation about anticipated and actual completion dates. If the treating clinician determined that adherence was inadequate (e.g. missed days/weeks) and recommended restarting or lengthening the duration of treatment we judged TPT completion against the revised date. If a participant refused TPT or ceased TPT early for any reason (including adverse effects) they were considered incomplete. We set the 'within one month' cut-off as a balance between misclassifying actual completes as incomplete (e.g. demonstrated adherence at 5.5 months and completed the final few weeks without further review) and misclassifying actual incompletes as complete (e.g. adherent at 3 months but did not fill repeat script to complete therapy). We consider this to be a conservative definition of treatment completion with the most likely source of misclassification coming from children who completed treatment but never returned for final clinical review (i.e. misclassified as 'incomplete'). By definition, children with TBI are asymptomatic and there are no clinical tests that can confirm whether TPT has been successful.

Secondary adherence outcomes included the number and proportion of appointments attended. Given this was a retrospective audit we did not attempt pill counts or drug levels to validate adherence.

We recorded a range of individual and family-level indicators that may influence TPT completion or help us understand the pathway of care. These included demographic characteristics (including age, sex, country of birth, preferred language and background risk of TB infection) and clinical variables–including TB testing results (Tuberculin skin testing (TST), Interferon Gamma Release Assay (IGRA), Chest X-ray), prescribed treatments and medication side effects, and other laboratory tests.

The minimum projected sample size required 95 children per group (Completed vs Incomplete treatment) to detect a 20-percentage point difference with 95% confidence and 80% power. We compared demographic and clinical variables between the completed and

incomplete groups using Pearson's chi-squared test (or Fisher's exact test where appropriate) for categorical variables and Student's t-tests for continuous variables. We used multiple logistic regression to determine the contribution of individual variables on the likelihood of treatment completion, with backward stepwise selection of candidate variables to construct the final model.

## Results

### Participant characteristics

We identified 402 participants with TBI, all of whom were offered TPT. One participant progressed to active TB whilst on TPT but was retained in the dataset for analysis. Of the 402 participants, 189 (47%) were female (**Table 1**). The average age was 8.1 years (standard deviation 5.03 years) with relatively even distribution of participants across all age groups (115 aged under 5 years, 111 aged 5 to 9 years, 176 aged 10 to 17 years). 313 (77%) children were born overseas and 249 (61.9%) had a preferred language other than English. The most common languages other than English were Burmese (18%), Persian (7.5%), Dinka (6.5%) and Arabic (5%). 165 children (41%) had a household contact with symptoms of confirmed TB as their highest exposure risk factor, while 54 (13%) had a non-household contact, 155 (39%) had lived in an endemic area and 24 (6%) had no known exposure risks but were tested for TB for other reasons (e.g. as part of routine refugee health arrival screening, or pre-immunosuppressive therapy workup).

### Treatment completion

Of the included 402 participants, 296 (74%) completed TPT to within 1 month of a target duration according to clinician documentation (Completed). The remainder (106, 26%) were defined as Incomplete, including ten participants were adherent until 1.5 months before their target duration but were then discharged without review.

**Predictors of treatment completion.** Age, sex and exposure risk level were not associated with TPT completion rates on univariate analysis (**Table 1**). TPT completion rates in older children and adolescents were similar to those of infants and young children of less than 5 years: 72% and 77% respectively. Completion rates were lower among those with English as preferred language than those with a preferred language other than English (66% vs 80%, p<0.001) and among those born in Australia compared to overseas (65% vs 76%, p = 0.018). Completion rates varied depending on where children were referred from, with highest treatment completion among children referred by their family doctor (82.5%) and lowest among children referred by the state TB public health unit following TB contact (62%). Completion rates were higher among those who received migrant screening (85% vs 65%, p<0.001) or treatment for other conditions (81% vs 64%, p<0.001) compared to those who did not. Completion rates were lower among those with side effects than without (47% vs 75%, p = 0.016), and similar regardless of medication change.

Multivariate logistic regression analysis showed that TPT completion was more likely among those who had refugee or migrant screening performed (OR 2.31, 95%CI 1.34–4.00) or received treatment for other medical conditions (OR 1.67 95%CI 1.0–2.85) compared to those who did not (Table 2, full model with all candidate variables in Table 2 in S1 Text). TPT completion was less likely among those who experienced side effects (OR 0.32, 95%CI 0.11–0.94). We included preferred language in the final model as it improved model fit even though it did not reach statistical significance, with a trend toward more likely completion among those with a preferred language other than English than those who primarily spoke English (OR

**Table 1. Characteristics of children diagnosed with latent tuberculosis infection at the Royal Children's Hospital, Melbourne, 2010–2016, with comparison on the basis of completion of TPT.**

| Demographics | Total (%) N = 402 | Completion Rate | Completed (%) N = 295 | Incomplete (%) N = 107 | P |
|---|---|---|---|---|---|
| Infant (<1 yr) | 20 (5.0%) | 60% | 12(4.1%) | 8(7.5%) | 0.132 |
| Young Child (1–4 yrs) | 95(23.6%) | 82% | 77(26.1%) | 18 (16.8%) | - |
| Older Child (5–9 yrs) | 111 (27.6%) | 69% | 77(26.1%) | 34 (31.8%) | - |
| Adolescent (10–17 yrs) | 176 (43.8%) | 73% | 129(43.7%) | 47 (43.9%) | - |
| **Sex** | | | | | |
| Male | 213(53.0%) | 76% | 159(54.0%) | 54 (50.4%) | 0.542 |
| Female | 189 (47.0%) | 71% | 136(46.1%) | 53 (49.5%) | - |
| **Country of birth** | | | | | |
| Australia | 91 (22.6%) | 65% | 58(19.7%) | 33 (30.8%) | 0.018 |
| Other | 311 (77.4%) | 76% | 237(80.3%) | 74 (69.2%) | - |
| **Preferred language** | | | | | |
| English | 146 (36.3%) | 66% | 94(31.9%) | 52 (48.6%) | <0.001 |
| Other | 249 (61.9%) | 80% | 199(67.5%) | 50 (46.7%) | - |
| Not recorded | 7 (1.7%) | 29% | 2 (0.7%) | 5 (4.7%) | - |
| **Highest risk factor** | | | | | |
| Household contact with symptoms or TB | 165 (41.0%) | 68% | 112(37.8%) | 52(50.0%) | 0.163 |
| Non-household contact with symptoms or TB | 54 (13.4%) | 78% | 42(14.2%) | 12 (11.3%) | - |
| Lived in endemic area | 155 (38.6%) | 79% | 122(41.4%) | 33 (30.8%) | - |
| Travelled in endemic area | 4 (1.0%) | 50% | 2 (0.7%) | 2 (1.9%) | - |
| No documented exposure risks | 24 (6.0%) | 75% | 18(6.1%) | 6 (5.7%) | - |
| **TPT regimen** | | | | | |
| 6H | 377 (93.8%) | 75% | 283 (95.6%) | 95 (88.7%) | 0.001 |
| Other | 21 (5.2%) | 62% | 13(4.4%) | 8 (7.6%) | - |
| Not recorded | 4 (1%) | 0% | 0(0%) | 4 (3.8%) | - |
| **Medication changed** | | | | | |
| Yes | 11(2.7%) | 73% | 8(2.7%) | 3 (2.8%) | 1 |
| No | 391 (97.3%) | 74% | 288 (97.3%) | 103 (97.2%) | - |
| **Side effects** | | | | | |
| Yes | 15 (3.7%) | 47% | 7(2.4%) | 8 (7.6%) | 0.016 |
| No | 387 (96.3%) | 75% | 286 (97.6%) | 98 (92.5%) | - |

P-values reflect test of difference in proportions comparing Complete and Incomplete groups using Pearson's chi-squared test. 6H = six-month daily isoniazid; TPT = Tuberculosis preventive therapy

1.45, 95%CI 0.88–2.39). In sensitivity analysis we substituted country of birth for preferred language, finding weaker association (OR 1.06, 95%CI 0.59–1.91).

## Pathway of care

Most participants were referred from a General Practitioner (GP) (44%) or the state TB Public Health Unit as contacts of an active TB case (35%). A significant minority were referred for migration related health screening (10.5%) and the remainder from immigration detention health service providers and internal hospital pathways (e.g. follow up from Emergency or admission).

**Fig 1** shows the pathway of care, TPT recommendations and clinical outcomes for participants. Most participants were directed into our specialist "TB clinic" (62%), which provides clinical care and advice on child TB at a state-wide level, or the general paediatric "Immigrant

**Table 2. Factors associated with completion of TB preventive therapy on multivariate analysis.**

| Variable | Final model | |
|---|---|---|
| | OR | 95% CI |
| Other language (vs English) | 1.45 | 0.88–2.39 |
| Refugee health screen | 2.31 | 1.34–4.00 |
| Other conditions treated | 1.67 | 1.0–2.85 |
| Medication side effects | 0.32 | 0.11–0.94 |

OR = Odds ratio; CI = Confidence interval; RCH = Royal Children's Hospital; TB = tuberculosis; TPT = TB preventive therapy.

Other candidate variables included: age; sex; country of birth; referral source; clinic attended; level of TB exposure risk; change of clinic; TPT regimen; target duration; change of medication; TB testing in community; TB testing at RCH. See Table 2 in S1 Text for details.

Health clinic" (28%), which serves children of refugee and asylum seeker background. A minority attended Infectious Diseases (ID) clinics (7%) and General Paediatrics clinics (1%).

Patients attended a median of four (IQR 3–6) appointments during their episode of TBI care, with a median appointment attendance rate of 83% (IQR: 67–100%). Most children (282, 70%) had already had TST or IGRA performed before attending RCH, and half of the children (224, 56%) had repeat or further testing performed at RCH (Table 3 in S1 Text). TST results were available for 346 (86%) children, with 271 (78%) positive. IGRA results were available for 204 (51%) children, with 111 (54%) positive and 4 (2%) indeterminate. Of 148 patients who had both TST and IGRA, 85 (57%) results were discordant–usually TST-positive IGRA-negative

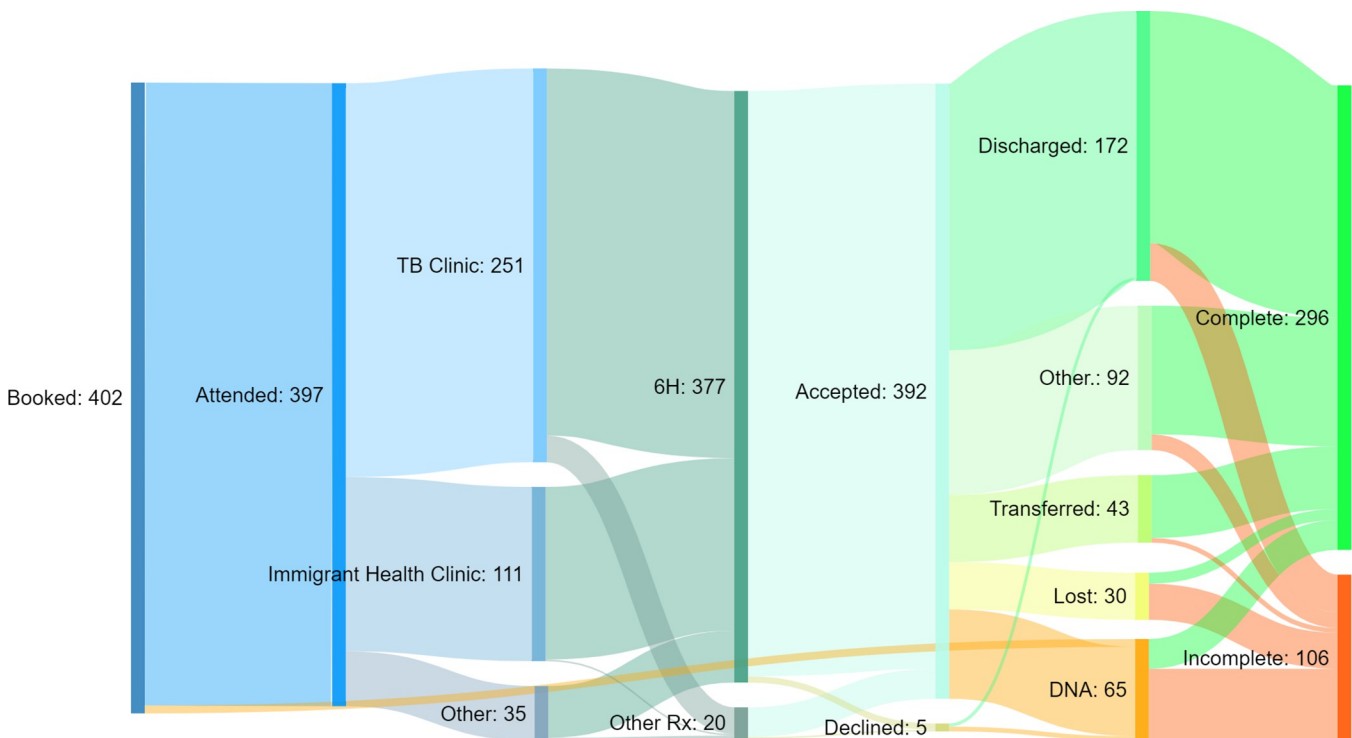

**Fig 1. Pathway of care and treatment completion outcomes for 402 children (aged <18 years) with latent tuberculosis infection (TBI) at the Royal Children's Hospital, Melbourne, 2010–2016.** Numbers correspond to the coloured column to the right, indicating the number of participants at various care-points.

(69, 47%) (Table 4 in S1 Text). Three-quarters (288, 72%) of children had other testing performed, most commonly relating to immigrant health arrival screening (Table 5 in S1 Text).

Almost all children (377, 94%) were prescribed a 6-month course of isoniazid (6H) as TPT. Other regimens included: isoniazid monotherapy for 9 months (9H), rifampicin monotherapy for 4 months (4R), and various combination therapies. Most patients and families who were recommended TPT accepted (392, 99%), with only five declining (**Fig 1**).

Three-quarters of children were either discharged having completed care (172, 43%), transferred to another provider to complete care (43, 11%), or continued seeing other RCH services for non-TB care (92, 23%). One-quarter did not attend their last appointment (65, 16%) or were lost to follow-up through administrative error (i.e. not rebooked when they should have been) (30, 7%).

## Discussion

We found that 74% of children and adolescents eligible for TPT completed treatment. This finding compares favourably to completion rates in a recent meta-analysis that included data from 13 high-income, low-incidence countries of uptake and completion of a range of TPT options in migrants, mainly adults, with TBI [10]. Pooled estimates found 74% completion for migrants who initiated TPT but that only 52% of those with evidence of TBI both initiated and completed TPT, with many drop-offs noted along the cascade of care [10]. Most children who develop TB in Australia are either recent migrants or born into a family of recent migrants [11]. We did not find significant differences in completion rates across age-disaggregated groups from infants to adolescents. This is an important observation given that WHO TPT recommendations for contact management in TB-endemic settings now include HIV-negative older children and adolescents, whereas the previous focus was on young child contacts and people living with HIV [3].

While overall completion rates were reasonable, we did identify opportunities for improved TPT completion, several which have been implemented recently. Reported barriers to TPT adherence and completion in the literature include adverse effects, competing priorities, transport or other access difficulties, pill burden, limited social support, high cost, poor patient-clinician relationship, administrative failures, and stigma or disease-related shame [12–15]. Facilitating factors for TPT adherence include strong individual motivation, good patient education, practical incentives (e.g. bus tickets), health service outreach, and shorter and simpler regimens [12, 15, 16]. Many of these same factors are identified in the much larger literature on TB treatment adherence, where strategies such as patient education and counselling, incentives, reminders, psychological interventions, and digital health technologies can improve patient outcomes [15].

Our results reveal some similarities and some differences to these adherence and completion factors. Like others, we found that experiencing adverse effects reduced the likelihood of treatment completion (OR 0.32, 95% CI 0.11–0.94), although our discontinuation rate due to adverse effects was comparably low (8/402, 2%) [10, 13, 16]. Similarly, we found a high proportion of children (23%) who were discharged after failed attendances or lost to follow up due to clerical error, a significant contributor to defaulting that is clearly amenable to intervention [15].

Our finding of higher TPT completion rates for those who have had refugee health screening or management of other conditions was less expected and could be due a combination of factors. These patients typically had more visits, more holistic health care (longer visits to build rapport and trust and complete a thorough evaluation), and likely benefitted from more intensive support from clinical nurse managers (e.g. reminders, rebooking) in the immigrant health

team. This would be consistent with studies showing the benefits of integrating and co-locating health services for newly arrived refugees and other studies showing the benefits of holistic approaches to care more broadly [17, 18]. However, it is also possible that certain socio-cultural factors (e.g. deference to medical personnel or authority figures), past experiences (e.g. greater familiarity with TB and its consequences), or other individual factors also contributed.

We may have expected to find lower completion rates among those with a preferred language other than English, given the additional challenges to communication, health service access, and higher frequency of economic and transport difficulties among recent migrants. However, completion rates were higher among this population than those with English as preferred language (67% vs 47% p = <0.001) with no clear association after adjusting for refugee health status. Interpreter use is clearly important and we were concerned about underutilisation in our service, as has been observed elsewhere [19]. However, documentation of interpreter use in our clinical records was inadequate to explore this further.

Since conducting this study our service has initiated changes to address some of these key barriers. We have built on our centralised intake for TB-related referrals, with a dedicated administrator to coordinate referrals and follow-up and a paediatric TB fellow to support service improvement. All TB referrals (and ward follow up) are now directed to the weekly TB clinic (or Immigrant health clinic for refugees) which is also attended by a public health nurse consultant from the state TB Program (who are responsible for home-based care and treatment support of index cases and their contacts) [20]. We have used the new electronic medical records system to streamline approaches to care–including templates and order sets to make assessment, management, education, and follow-up easier.

Our study included data from a time when the standard TPT was at least six months of daily isoniazid. Given that this was used in the majority, it was not possible to compare completion between alternative regimens. However, we have recently changed guidelines and practice by the introduction of shorter TPT regimens, [4] including 4 months of rifampicin for older children and a dispersible, fixed-dose combination of isoniazid-rifampicin for 3 months in younger children. These shorter TPT regimens are associated with improved adherence and fewer side-effects and are included in updated WHO recommendations [3, 7]. In eligible young child contacts, high uptake and completion rates of over 90% were reported following the introduction of the 3-month regimen of the child-friendly fixed-dose formulation in urban clinics in a number of African countries, along with very few reported side-effects [21]. Similarly, the meta-analysis of largely adult migrants with TBI, which included data from Australia, noted higher initiation and completion rates with shorter regimens compared to 6 to 9 months of isoniazid preventive treatment [10].

The final change has been a move towards decentralisation of services, something that has been influenced and accelerated by responses to the COVID19 pandemic. Until recently, almost all children in Victoria requiring TPT attended the one urban-based hospital, RCH. Yet, such children are usually well with no co-morbidities and so outpatient care with TPT could be provided closer to home. Evidence from TB programs globally suggests that decentralisation of services can improve detection, treatment and prevention of TB in children [22, 23]. Just prior to the COVID pandemic, and accelerated further since, we have worked with colleagues to strengthen TB services for children in additional locations. We have also sought to provide our specialist input more flexibly, particularly through telehealth with families and primary care providers. In practice, this typically means children requiring initial assessment and TPT initiation will be seen face-to-face with their caregivers and a skilled TB doctor (at RCH or elsewhere), with ongoing treatment support and assessment often provided by telehealth.

## Limitations

Our study is a retrospective audit, and therefore relies upon the accuracy of documentation by administrative staff and clinicians. We used a conservative clinical definition of treatment completion in our analysis that required clinical confirmation within 1 month of the target treatment duration (eg. at 5 months for those recommended 6 months of isoniazid). This clinical approach to adherence is consistent with most other studies and there may have been some children who were given medications well in advance and completed therapy but were not reviewed again. No objective measure of adherence was undertaken such as pill counts or drug levels. Our audit may not have captured patients with TBI that attended other clinics and did not come up in diagnostic codes or pharmacy prescription records, but we expect that this number would be very small and would not significantly influence our findings. Finally, the study was performed in a high-resource, low-prevalence setting with a robust TB control program. Therefore, generalisability of the results to lower resource or higher prevalence settings is limited.

## Conclusion

The completion of TPT in children was influenced by individual, social and clinical factors. Children who had refugee or migrant screening performed, or were being treated for other conditions, were more likely to complete treatment, whereas children who experienced adverse medication side effects were less likely to complete treatment. The findings have informed efforts to improve our TB service, including the recent introduction of shorter TPT regimens.

## Supporting information

**S1 Checklist. STROBE statement—checklist of items that should be included in reports of observational studies.**
(PDF)

**S1 Text. Supplemental extended results document.**
(DOCX)

**S2 Text. Supplemental data analysis document.**
(TXT)

## Author Contributions

**Conceptualization:** Sarath Ranganathan, Hamish R. Graham.

**Data curation:** Sunjuri Sun, Saniya Kazi, Hamish R. Graham.

**Formal analysis:** Rebecca Helena Holmes, Stephen M. Graham, Hamish R. Graham.

**Investigation:** Stephen M. Graham.

**Methodology:** Sarath Ranganathan, Shidan Tosif, Hamish R. Graham.

**Project administration:** Hamish R. Graham.

**Supervision:** Stephen M. Graham, Hamish R. Graham.

**Writing – original draft:** Rebecca Helena Holmes.

**Writing – review & editing:** Rebecca Helena Holmes, Sunjuri Sun, Saniya Kazi, Sarath Ranganathan, Shidan Tosif, Stephen M. Graham, Hamish R. Graham.

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
