## [Decision Letter · Decision Letter 0]

15 Aug 2022

PONE-D-21-40361Management of tuberculosis infection in Victorian children: A retrospective clinical audit of factors affecting treatment completionPLOS ONE

Dear Dr. Graham,

Thank you for submitting your manuscript to PLOS ONE. After careful consideration, we feel that it has merit but does not fully meet PLOS ONE’s publication criteria as it currently stands. Therefore, we invite you to submit a revised version of the manuscript that addresses the points raised during the review process.

Please pay particular attention to responding to the concerns raised by Reviewer 1 regarding the classification of complete and incomplete groups, as well as their presentational recommendations.

We look forward to receiving your revised manuscript.

Kind regards,

Jamie Males

Editorial Office

PLOS ONE

Journal Requirements:

2. Thank you for stating the following in the Competing Interests section: "HG, SG, ST, SK, SR are employed by the Royal Children’s Hospital and work in the TB clinic."

Reviewers' comments:

Reviewer's Responses to Questions

**Comments to the Author**

1. Is the manuscript technically sound, and do the data support the conclusions?

Reviewer #1: Yes

Reviewer #2: Yes

2. Has the statistical analysis been performed appropriately and rigorously? 

Reviewer #1: Yes

Reviewer #2: Yes

3. Have the authors made all data underlying the findings in their manuscript fully available?

Reviewer #1: No

Reviewer #2: No

4. Is the manuscript presented in an intelligible fashion and written in standard English?

Reviewer #1: Yes

Reviewer #2: Yes

5. Review Comments to the Author

Reviewer #1: This is a well written and straight forward piece of research. The research team have examined completion of TB treatment in young people. It has been performed as a clinical quality improvement project for their service but I believe it has greater applicability to other services in higher income countries.

My queries relate to the classification of complete and incomplete groups. Is there potential for misclassification in these groups. The complete classification is discussed in limitations but this should be further up forward in the manuscript. Does 5 out of 6 months indicate a good clinical outcome of treatment? Can the authors include additional justification of the underpinnings of this choice? Also, the incomplete group are there any children in this group who are incomplete but who followed all clinical advice and so would be considered complete? that is a medical complication that interfered with treatment or other factors that interfered and may cause misclassification? I think that this can all be handled with some additional explanation in the text.

Also, Table 2 in the Supplemental section presents the multivariate analyses. I believe this would be better incorporated as part of the manuscript. The table can be abbreviated to fit in the manuscript, but I think this information is helpful for the reader to have available without searching the Supplemental material.

Ln104 in the manuscript mentions the previous 10 years but at Ln88 the manuscript mentions a period of 7 years between 2010 and 2016. Please check and correct if necessary

Supplementary information in Table 2 footer states "Odds ration" and should be corrected to ratio.

The short hand of 6H is used in Table 1 and in the footer of that table. Also appears in Figure 1. However, this is not described in text. It would be useful for the reader to be introduced to this while reading about isoniazid treatment in the text.

Reviewer #2: The authors should be commended for examining TB preventive therapy among children. They appropriately recognize the limitations of a retrospective audit but laudably describe the use of the results to improve patient care. This is implementation science and the manuscript could have benefited from using an implementation science lens, but this is not requisite. A few clarifications are needed however:

1. Line 72, Intro 2nd paragraph “in younger children with…” something is missing here

2. Line 80, Intro, last paragraph, “parental barriers” – what kind of parental factors? Were parental factors, in fact, assessed?

3. Line 83, Intro, last paragraph “more effective TPT” – compared to what?

4. Line 141-142, results “6% had no known exposure risks” – why were they tested?

5. Line 152 or line 210, results, “discharged without review” – kindly clarify how this happened?

6. Line 173, results typo: “that”

7. Line 296, discussion “conservative clinical definition” – on what is this clinical definition based?

8. Line 308 – individual, social, clinical factors – but earlier in the text the authors state “Individual, parental, institutional” – kindly be consistent

6. PLOS authors have the option to publish the peer review history of their article (what does this mean?). If published, this will include your full peer review and any attached files.

Reviewer #1: No

Reviewer #2: No

---

## [Author Response · Author response to Decision Letter 0]

22 Aug 2022

(See uploaded file)

Response to Reviewers and Editor

PONE-D-21-40361

Management of tuberculosis infection in Victorian children: A retrospective clinical audit of factors affecting treatment completion

Thank you for your Reviews and Editorial feedback, including request for Minor Revision. We have revised the manuscript according to your suggestions and providing the additional information requested – see details below.

Please find attached:

• A marked-up copy of our manuscript that highlights changes made to the original version.

• An unmarked version of your revised paper without tracked changes. 

• Updated supplemental material document

• New supplemental data analysis file

Journal Requirements:

Checked and updated formatting.

2. Thank you for stating the following in the Competing Interests section: "HG, SG, ST, SK, SR are employed by the Royal Children’s Hospital and work in the TB clinic."

Updated and included in Cover Letter as requested.

Our audit was conducted under the Royal Children's Hospital permissions for quality improvement activities and does not include permission to share individual patient data in an open access repository. We can share minimum data through data sharing agreements with individuals/institutions on request. We can also make the complete analysis file openly available for transparency on what we have done. 

Updated information on data sharing: Deidentified data summaries and analysis file are provided in supplemental material. Our audit was conducted under the Royal Children's Hospital permissions for quality improvement activities and does not include permission to share individual patient data in an open access repository. The data underlying the results presented in the study are available for researchers who meet the criteria for access to confidential data from The Royal Children's Hospital HREC (rch.ethics@rch.org.au; +61 393455044).

We have removed the ethics statement from additional information section and retained it in the Methods.

Formatting revised accordingly.

Checked.

Reviewers' comments:

Reviewer's Responses to Questions

Comments to the Author

1. Is the manuscript technically sound, and do the data support the conclusions?

Reviewer #1: Yes

Reviewer #2: Yes

2. Has the statistical analysis been performed appropriately and rigorously?

Reviewer #1: Yes

Reviewer #2: Yes

3. Have the authors made all data underlying the findings in their manuscript fully available?

Reviewer #1: No

Reviewer #2: No

4. Is the manuscript presented in an intelligible fashion and written in standard English?

Reviewer #1: Yes

Reviewer #2: Yes

5. Review Comments to the Author

Reviewer #1: This is a well written and straight forward piece of research. The research team have examined completion of TB treatment in young people. It has been performed as a clinical quality improvement project for their service but I believe it has greater applicability to other services in higher income countries.

Thank you for your kind comments and your helpful suggestions below. 

My queries relate to the classification of complete and incomplete groups. Is there potential for misclassification in these groups. The complete classification is discussed in limitations but this should be further up forward in the manuscript. Does 5 out of 6 months indicate a good clinical outcome of treatment? Can the authors include additional justification of the underpinnings of this choice? Also, the incomplete group are there any children in this group who are incomplete but who followed all clinical advice and so would be considered complete? that is a medical complication that interfered with treatment or other factors that interfered and may cause misclassification? I think that this can all be handled with some additional explanation in the text.

Thank you. We have revised the Methods to be clearer about the TPT completion definition and risk of misclassification. 

“The primary outcome was TPT completion, defined as completing pharmacological treatment to within 1 month of a target duration for the specified agent (for instance, at least 5 months of a 6-month isoniazid course), confirmed by the treating clinician. We determined this by identifying the recommended treatment and date of commencement, then checking all subsequent clinical encounters to find clinical documentation about anticipated and actual completion dates. If the treating clinician determined that adherence was inadequate (e.g. missed days/weeks) and recommended restarting or lengthening the duration of treatment we judged TPT completion against the revised date. If a participant refused TPT or ceased TPT early for any reason (including adverse effects) they were considered incomplete. We set the ‘within 1 month’ cut-off as a balance between misclassifying actual completes as incomplete (e.g. demonstrated adherence at 5.5 months and completed the final few weeks without further review) and misclassifying actual incompletes as complete (e.g. adherent at 3 months but did not fill repeat script to complete therapy). We consider this to be a conservative definition of treatment completion with the most likely source of misclassification coming from children who completed treatment but never returned for final clinical review (i.e. misclassified as ‘incomplete’). By definition, children with TBI are asymptomatic and there are no clinical tests that can confirm whether TPT has been successful.”

Also, Table 2 in the Supplemental section presents the multivariate analyses. I believe this would be better incorporated as part of the manuscript. The table can be abbreviated to fit in the manuscript, but I think this information is helpful for the reader to have available without searching the Supplemental material.

Thankyou for your suggestion. We have moved an abbreviated multivariate analysis Table into the main text and retained the full table (with all variables) in the supplemental material. 

Ln104 in the manuscript mentions the previous 10 years but at Ln88 the manuscript mentions a period of 7 years between 2010 and 2016. Please check and correct if necessary

Thank you. Corrected to past 7 years.

Supplementary information in Table 2 footer states "Odds ration" and should be corrected to ratio.

Corrected.

The short hand of 6H is used in Table 1 and in the footer of that table. Also appears in Figure 1. However, this is not described in text. It would be useful for the reader to be introduced to this while reading about isoniazid treatment in the text.

Added 6H abbreviation to the text.

Reviewer #2: The authors should be commended for examining TB preventive therapy among children. They appropriately recognize the limitations of a retrospective audit but laudably describe the use of the results to improve patient care. This is implementation science and the manuscript could have benefited from using an implementation science lens, but this is not requisite. A few clarifications are needed however:

1. Line 72, Intro 2nd paragraph “in younger children with…” something is missing here

Thank you. Deleted ‘with’.

2. Line 80, Intro, last paragraph, “parental barriers” – what kind of parental factors? Were parental factors, in fact, assessed?

Changed to individual, social, and clinical factors for consistency with Discussion/Conclusion.

3. Line 83, Intro, last paragraph “more effective TPT” – compared to what?

Compared to currently. Revised to state “… some of these factors may be modifiable by service providers to deliver TPT more effectively”.

4. Line 141-142, results “6% had no known exposure risks” – why were they tested?

These children were typically refugee-background from countries not considered high risk for TB but screened for TB as part of routine recommended refugee health arrival screening. Or TB testing as part of work up before starting immunosuppressive therapy. Added this to text.

5. Line 152 or line 210, results, “discharged without review” – kindly clarify how this happened?

Treating clinicians differed in how frequently they would review children and at what stage they would discharge patients. We remark on these in particular because it is likely that the treating clinician was happy that they would complete treatment without further review.

6. Line 173, results typo: “that”

Corrected.

7. Line 296, discussion “conservative clinical definition” – on what is this clinical definition based?

Full description of the TPT completion definition added in Methods.

The primary outcome was TPT completion, defined as completing pharmacological treatment to within 1 month of a target duration for the specified agent (for instance, at least 5 months of a 6-month isoniazid course), confirmed by the treating clinician. We determined this by identifying the recommended treatment and date of commencement, then checking all subsequent clinical encounters to find clinical documentation about anticipated and actual completion dates. If the treating clinician determined that adherence was inadequate (e.g. missed days/weeks) and recommended restarting or lengthening the duration of treatment we judged TPT completion against the revised date. If a participant refused TPT or ceased TPT early for any reason (including adverse effects) they were considered incomplete. We set the ‘within 1 month’ cut-off as a balance between misclassifying actual completes as incomplete (e.g. demonstrated adherence at 5.5 months and completed the final few weeks without further review) and misclassifying actual incompletes as complete (e.g. adherent at 3 months but did not fill repeat script to complete therapy). We consider this to be a conservative definition of treatment completion with the most likely source of misclassification coming from children who completed treatment but never returned for final clinical review (i.e. misclassified as ‘incomplete’). By definition, children with TBI are asymptomatic and there are no clinical tests that can confirm whether TPT has been successful. 

8. Line 308 – individual, social, clinical factors – but earlier in the text the authors state “Individual, parental, institutional” – kindly be consistent

Changed to individual, social, and clinical factors for consistency.

---

## [Decision Letter · Decision Letter 1]

26 Sep 2022

Management of tuberculosis infection in Victorian children: A retrospective clinical audit of factors affecting treatment completion

PONE-D-21-40361R1

Dear Dr. Graham

We’re pleased to inform you that your manuscript has been judged scientifically suitable for publication and will be formally accepted for publication once it meets all outstanding technical requirements.

Kind regards,

Margaret Williams, Ph.D

Academic Editor

PLOS ONE

Additional Editor Comments (optional):

Reviewers' comments:

Reviewer's Responses to Questions

**Comments to the Author**

1. If the authors have adequately addressed your comments raised in a previous round of review and you feel that this manuscript is now acceptable for publication, you may indicate that here to bypass the “Comments to the Author” section, enter your conflict of interest statement in the “Confidential to Editor” section, and submit your "Accept" recommendation.

Reviewer #1: All comments have been addressed

Reviewer #2: All comments have been addressed

2. Is the manuscript technically sound, and do the data support the conclusions?

Reviewer #1: Yes

Reviewer #2: Yes

3. Has the statistical analysis been performed appropriately and rigorously? 

Reviewer #1: Yes

Reviewer #2: Yes

4. Have the authors made all data underlying the findings in their manuscript fully available?

Reviewer #1: Yes

Reviewer #2: No

5. Is the manuscript presented in an intelligible fashion and written in standard English?

Reviewer #1: Yes

Reviewer #2: Yes

6. Review Comments to the Author

Reviewer #1: Thank you for addressing both of the reviewer's comments. The manuscript is improved by adding some points of clarification.

Reviewer #2: Thank you for your attention to the initial review. All of my queries have been addressed. This paper adds to the data regarding implementation of TB preventive therapy among children.

7. PLOS authors have the option to publish the peer review history of their article (what does this mean?). If published, this will include your full peer review and any attached files.

Reviewer #1: No

Reviewer #2: No

---

## [Editor Report · Acceptance letter]

3 Oct 2022

PONE-D-21-40361R1 

Management of tuberculosis infection in Victorian children: A retrospective clinical audit of factors affecting treatment completion 

Dear Dr. Graham:

I'm pleased to inform you that your manuscript has been deemed suitable for publication in PLOS ONE. Congratulations! Your manuscript is now with our production department. 

Kind regards, 

on behalf of

Professor Margaret Williams 

Academic Editor

PLOS ONE